# Grafted Microparticles Based on Glycidyl Methacrylate, Hydroxyethyl Methacrylate and Sodium Hyaluronate: Synthesis, Characterization, Adsorption and Release Studies of Metronidazole

**DOI:** 10.3390/polym14194151

**Published:** 2022-10-03

**Authors:** Aurica Ionela Gugoasa, Stefania Racovita, Silvia Vasiliu, Marcel Popa

**Affiliations:** 1Departament of Natural and Synthetic Polymers, Faculty of Chemical Engineering and Environmental Protection, “Gheorghe Asahi” Technical University of Iasi, Prof. Dr. Docent Dimitrie Mangeron Street No. 73, 700050 Iasi, Romania; 2“Petru Poni” Institute of Macromolecular Chemistry, Grigore Ghica Voda Alley No. 41A, 700487 Iasi, Romania; 3Academy of Romanian Scientists, Splaiul Independentei Street No. 54, 050085 Bucharest, Romania

**Keywords:** grafted microparticles, sodium hyaluronate, crosslinked microparticles, metronidazole adsorption, adsorption isotherm, adsorption kinetics, metronidazole release

## Abstract

Three types of precursor microparticles based on glycidyl methacrylate, hydroxyethyl methacrylate and one of the following three crosslinking agents (mono-, di- or triethylene glycol dimethacrylate) were prepared using the suspension polymerization technique. The precursor microparticles were subsequently used to obtain three types of hybrid microparticles. Their synthesis took place by grafting sodium hyaluronate, in a basic medium, to the epoxy groups located on the surface of the precursor microparticles. Both types of the microparticles were characterized by: FTIR spectroscopy, epoxy groups content, thermogravimetric analysis, dimensional analysis, grafting degree of sodium hyaluronate, SEM and AFM analyses, and specific parameters of porous structures (specific surface area, pore volume, porosity). The results showed that the hybrid microparticles present higher specific surface areas, higher swelling capacities as well as higher adsorption capacities of antimicrobial drugs (metronidazole). To examine the interactions between metronidazole and the precursor/hybrid microparticles the adsorption equilibrium, kinetic and thermodynamic studies were carried out. Thus, it was determined the performance of the polymer systems in order to select a polymer–drug system with a high efficiency. The release kinetics reflect that the release mechanism of metronidazole in the case of hybrid microparticles is a complex mechanism characteristic of anomalous or non-Fickian diffusion.

## 1. Introduction

The dynamics of medical research around the world are being directed towards developing new materials to solve and remedy various health problems. In recent years, dentistry, a part of the medical field, has been seeking to solve a series of problems related to the prevention of dental caries, periodontal disease and the treatment of bone tissue loss [1]. It is known that periodontitis (periodontal disease) is a chronic inflammatory condition of the gingiva and the tissues supporting teeth on their arches (gingiva, periodontal ligament, alveolar bone) [2,3]. The World Health Organization considers periodontal disease as one of the most common diseases of the oral cavity, statistically affecting three quarters of the world’s population.

The treatment of periodontal disease is complex and generally aims to slow down the evolution of the disease. In the early stages of the disease, local treatment is instituted [3], which is generally antimicrobial and consists of: scaling, which helps remove tartar and bacteria from the tooth surface and under the gingiva; and the use of antibiotics.

The development of polymer science and polymer characterization methods has led to the use of natural and synthetic polymers to produce polymeric materials in the form of microparticles to improve the prevention, diagnosis and treatment of injured tissues.

Among the natural polymers, chitosan, pectin, alginate, starch and dextran are used to prepare microparticles loaded with different drugs to treat periodontal diseases [4,5,6]. Another important polysaccharide, hyaluronic acid, which is used in the cosmetic industry, prepares the scaffold for tissue engineering for drug delivery, disease diagnoses and biomedical imaging; it represents an ideal candidate to obtain polymeric materials for the dental field [7,8,9,10]. The use of polymeric microparticles in dentistry has been increasing due to the excellent properties of both polymeric materials (good surface, biological and mechanical properties, low manufacturing cost, simple synthesis methods) as well as of microparticles (high surface/volume ratio, good accessibility to the site of action due to their size, stability in biological fluids, drug transport capacity, decreased frequency and intensity of adverse effects) [11,12]. Polymer-based microparticles can be achieved through several processes, the most important of which is polymerization. Among the polymerization methods used to obtain spherical microparticles (bulk polymerization, precipitation polymerization, dispersion polymerization, emulsion polymerization, etc.), aqueous suspension polymerization can be used because of the technical and economic advantages it confers. Since in suspension polymerization a small number of reagents and simple reaction equipment are used, the purification of the resulting products takes place by simple methods and the cost of obtaining the products is low; thus, it can be concluded that this method is suitable for obtaining polymeric materials with special properties that can be successfully applied both in the field of environmental protection and in the medical or pharmaceutical fields [13,14,15].

The main aim of this work is to develop polymeric materials in the form of porous microparticles with special architectures that are obtained by combining simple, economical and environmentally friendly methods that combine the properties of synthetic polymers (chemical, mechanical and thermal stability) with those of natural polymers (biocompatibility, biodegradability, non-toxicity) and have potential applications in the dental field. The choice of sodium hyaluronate, a derivative of hyaluronic acid, as a natural component in the production of the hybrid microparticles was based on the following considerations: hyaluronic acid is an essential component of the periodontal ligament matrix; it plays important roles in cell migration, adhesion and differentiation by binding proteins and cell receptors; and it has been studied as a metabolite or diagnostic marker of inflammation in gingival crevicular fluid [16,17]. Thus, by grafting hyaluronic acid onto the surface of precursor microparticles, it was intended to obtain a system for the delivery of chemotherapeutic agents (antimicrobial and anti-inflammatory drugs) to treat dental diseases. For obtaining polymer–drug systems, the biological active principle chosen was metronidazole, which is a bactericide that can be administered orally together with other antibiotics with an increased efficacy in treating periodontal disease.

## 2. Materials and Methods

### 2.1. Materials

All reagents used in the preparation and characterization of the precursor/hybrid microparticles were purchased from Sigma-Aldrich (Schnelldorf, Germany). Glycidyl methacrylate (GMA) and hydroxyethyl methacrylate (HEMA) were distilled before use under reduced pressure to remove the inhibitor. Dimethacrylic monomers [ethylene glycol dimethacrylate (EGDMA), diethylene glycol dimethacrylate (DEGDMA) and triethylene glycol dimethacrylate (TEGDMA)], the initiator [benzoyl peroxide (BOP)], porogenic agent [butyl acetate], poly(vinyl alchohol) (PVA, M_w_ = 67,000 g/mol, degree of hydrolysis, 88%), gelatine, NaCl, sodium hyaluronate (HA), HBr, NaOH, glacial acetic acid, Crystal violet, n-heptane, methanol, ethanol and metronidazole (M_w_ = 171.064 g/mol) are of analytical grade and were used as received.

### 2.2. Methods

#### 2.2.1. Synthesis of Precursor Microparticles

The precursor microparticles based on GMA, HEMA and EGDMA, DEGDMA or TEGDMA denoted AE, AD and AT were synthesized by the suspension polymerization technique in a 250 cm^3^ cylindrical reactor fitted with mechanical stirrer, thermometer and reflux condenser. The reaction mixture is formed by two phases:Aqueous phase containing a polymeric stabilizer (2 wt% mixture of PVA and gelatine) and NaCl (3 wt%);Organic phase is formed by GMA (70% mol), HEMA (20% mol), crosslinking agents (10% mol of EGDMA, DEGDMA or TEGDMA), BOP and butyl acetate at a dilution of D = 0.6.

The copolymerization reaction was conducted under a nitrogen atmosphere for 8 h at 78 °C and 1 h at 90 °C with a stirring rate of 400 rpm. After the copolymerization reactions were completed, the precursor microparticles were separated by decantation and washed with hot water. Then, to remove traces of residual monomers and porogenic agent, the precursor microparticles were extracted with methanol or ethanol in a Soxhlet apparatus.

#### 2.2.2. Synthesis of Hybrid Microparticles

The hybrid microparticles denoted AEHA, ADHA and ATHA were prepared by grafting the sodium hyaluronate to the epoxy groups situated on the surface of the precursor microparticles. A known quantity of precursor microparticles was immersed in a solution of various concentrations of sodium hyaluronate (0.2–1%) obtained by dissolving the polysaccharide in water with the addition of NaOH (pH = 7–10). The reaction took place at temperatures between 30 and 60 °C over a period of time ranging between 2 and 10 h. After the aforementioned time, the hybrid microparticles were filtered off, washed with water to remove unreacted polysaccharide and NaOH and then dried under a vacuum at 40 °C for 24 h.

#### 2.2.3. Infrared Spectroscopy

The precursor/hybrid microparticles were characterized by FTIR spectroscopy using a Bruker Vertex FT-IR Spectrometer at a resolution of 2 cm^−1^ in the range of 4000–400 cm^1^ by KBr pellet technique. In order to obtain FT-IR spectra, 0.03 g the precursor/hybrid microparticles or HA were mixed and ground with potassium bromide.

#### 2.2.4. Epoxy Group Content

The epoxy group content was determined by ASTM D1652 (standard test method for epoxy content of epoxy resins) [18]. This method consisted of direct titration with a standard solution of HBr in glacial acetic acid. Experimental values of epoxide equivalent weight (*EEW*) were determined using the following equation:(1)$$EEW=\frac{100\cdot w}{N\cdot V}\;\left(\mathrm{mol}\cdot g^{-1}\right)\;$$
where *w*—grams of precursor/hybrid microparticles; *N*—normality of the HBr in acetic acid (mol·L^−1^); and *V*—volume of HBr solution used for titration (mL).

#### 2.2.5. Thermogravimetric Analysis (TGA)

The thermal behavior of the precursor/hybrid microparticles (4 mg of sample) was performed at a heating rate of 10 °C·min^−1^ in nitrogen atmosphere, using a Mettler Toledo TGA 851 Derivatograph.

#### 2.2.6. Scanning Electron Microscopy (SEM)

The surface morphology of the precursor/hybrid microparticles was analyzed with a Quanta 200 environmental scanning electron microscope at 25 kV.

#### 2.2.7. Atomic Force Microscopy (AFM)

AFM images for the precursor/hybrid microparticles were performed using a Scanning Probe Microscope Solver Pro-M platform (NT-MDT, Moscow, Russia) with a rectangular silicon cantilever NSG 10 and 203 kHz oscillation frequency, in air at ambient temperature (23 °C). The latest version of NT-MDT NOVA software was used for the analysis and calculation of the microparticle surface characteristic parameters (arithmetic mean deviation of the surface (*S_a_*); root-mean-square deviation of the surface (*S_q_*); surface skewness (*S_sk_*); and surface kurtosis (*S_ku_*)) and AFM imaging. The equations of the three-dimensional roughness parameters used in the current study are presented below [19]:(2)$$\;S_{a}=\frac{1}{MN}\overset{N}{\underset{j=1}{\sum }}\overset{M}{\underset{i=1}{\sum }}\left|z\left(x_{i},y_{j}\;\right)-\bar{z}\right|\;$$
(3)$$\;S_{q}={\left[\frac{1}{MN}\overset{N}{\underset{j=1}{\sum }}\overset{M}{\underset{i=1}{\sum }}{\left|z\left(x_{i},y_{j}\right)-\bar{z}\right|}^{2}\right]}^{1/2}\;$$
(4)$$\;S_{sk}=\frac{1}{MNS_{q}^{3}}\overset{N}{\underset{j=1}{\sum }}\overset{M}{\underset{i=1}{\sum }}{\left|z\left(x_{i},y_{j}\right)-\bar{z}\right|}^{3}\;$$
(5)$$\;S_{ku}=\frac{1}{MNS_{q}^{4}}\overset{N}{\underset{j=1}{\sum }}\overset{M}{\underset{i=1}{\sum }}{\left|z\left(x_{i},y_{i}\right)-\bar{z}\right|}^{4}\;$$
where *N*—number of points along of scan line; *M*—number of lines; *S_q_*—the root mean square roughness; and *z*—the height of each point of coordinates *x_i_* and *x_j_*.

Additionally, shape and elongation factors are two important parameters in pore structure analysis and can be calculated with the following equations:(6)$$\;f_{shape}=\frac{4\cdot \pi \cdot A}{1.064\cdot P^{2}}$$
(7)$$\;f_{elongation}=\frac{D_{min}}{D_{max}}$$
where *A*—pore area; *P*—pore perimeter; *D_min_*—minimum Feret diameter; and *D_max_*—maximum Feret diameter.

#### 2.2.8. Dimensional Analysis of Precursor/Hybrid Microparticles

The number average diameter (D) was obtained using a WingSALD 7001 laser diffraction particle size analyzer (UK). The measurements were performed by suspending the precursor microparticles or hybrid microparticles in methanol (non-solvent). The experimental data were recorded and processed using WingSALD software.

#### 2.2.9. Specific Parameters for the Characterization of the Morphology of Porous Structure

The morphology of the porous structure of the precursor/hybrid microparticles can be characterized using the following parameters: porosity (*P*, *%*), pore volume (*PV*, mL·g^−1^) and specific surface area (*S_sp_*, m^2^·g^−1^). The pore volume and the porosity of the precursor/hybrid microparticles were calculated as follows:(8)$$PV=\frac{1}{\rho_{ap}}-\frac{1}{\rho_{sp}}$$
(9)$$\%P=100\cdot \left(1-\frac{\rho_{ap}}{\rho_{sp}}\right)$$
where *ρ_ap_*—apparent density (g·cm^−3^); and *ρ_sp_*—skeletal density (g·cm^−3^).

The apparent and skeletal densities of precursor/hybrid microparticles were measured by pycnometric methods with mercury and n-heptane, respectively [20] and were calculated with the following equations:(10)$$\;\rho_{ap}=\frac{m_{1}}{V_{P}-\left(m_{3}-m_{2}\right)/\rho_{Hg}}$$
(11)$$\;\rho_{sp}=\frac{m_{1}}{V_{f}-\left(m_{s}-m_{4}\right)/h}\;$$
where *m_1_*—mass of the sample (precursor/hybrid microparticles) (g); *V_P_*—volume of the pycnometer (cm^3^); *V_f_*—volume of the volumetric flask (cm^3^); *m_2_*—mass of the pycnometer with the sample (g); *m_3_*—mass of the pycnometer with mercury and the sample (g); *ρ_Hg_*—density of mercury (g·cm^−3^); *m_4_*—mass of the volumetric flask with the sample (g); *m_S_*—mass of the volumetric flask with the sample and n-heptane(g); and *h*—density of n-heptane (g·cm^−3^).

The specific surface area was determined by dynamic vapor sorption using the Brunauer, Emmet and Teller (BET) method [21] and the sorption–desorption curves recorded for the precursor/hybrid microparticles. Sorption–desorption isotherms were registered using the fully automated gravimetric analyzer IGAsorp produced by Hiden Analytical, Warrington (UK).

#### 2.2.10. Swelling Studies

The swelling capacities of the precursor/hybrid microparticles were determined in aqueous solution at pH 1.2 and 5.5, respectively, using the gravimetric method. A known amount of dried precursor/hybrid microparticles (0.2 g) was immersed in 10 mL aqueous solution at 25 °C. At a certain period of time ranging between 10 and 1440 min, the precursor/hybrid microparticles were removed, centrifuged at 500 rpm for 10 min and weighed. The swelling capacity of the precursor/hybrid microparticles was calculated using the following equation:(12)$$\;S_{w}\;\left(\%\right)=\frac{w_{S}-w_{d}}{w_{d}}\;\cdot \;100$$
where *w_S_*—amount of swollen precursor/hybrid microparticles (g); and *w_d_*—amount of dry precursor/hybrid microparticles (g).

#### 2.2.11. Bach Adsorption Studies

The adsorption of metronidazole on precursor/hybrid microparticles was investigated in a batch system. Metronidazole adsorption was realized as follows: 0.1 g of precursor/hybrid microparticles of known moisture were introduced in 50 mL conical flasks filled with 10 mL metronidazole solution with various initial concentrations (0.25–1 mg·mL^−1^). The conical flasks were placed in a thermostatic shaker bath (Memmert M00/M01, Germany) and shaken at 180 rpm and 25, 30 and 40 °C for different periods of time ranging from 10 to 1440 min. After the specified period of time, the precursor/hybrid microparticles were removed quantitatively from the metronidazole solution by centrifugation at 1000 rpm for 10 min. The concentration of metronidazole in the supernatant solution before and after adsorption was determined using a UV–VIS spectrophotometer (UV–VIS SPEKOL 1300, Analytik Jena, Jena, Germany) at a wavelength of 277 nm based on the calibration curve obtained with various drug solutions of known conditions. The amounts of metronidazole at equilibrium, *q_e_* (mg·g^−1^), and at any time, *q_t_* (mg·g^−1^), were calculated from the following equations:(13)$$\;q_{e}=\frac{\left(C_{0}-C_{e}\right)\cdot V}{w}\;$$
(14)$$\;q_{t}=\frac{\left(C_{0}-C_{t}\right)\cdot V}{w}$$
where *C*_0_—initial concentration of metronidazole solution (mg·g^−1^); *C_e_*—concentration of metronidazole at equilibrium (mg·g^−1^); *C_t_*—concentration of metronidazole at any time (mg·g^−1^); *V*—volume of drug solution (L); and *w*—amount of precursor/hybrid microparticles (g).

#### 2.2.12. Drug Release Studies

In vitro drug release studies were carried out as follows: 100 mg of the drug-microparticle systems were introduced in 10 mL of buffer solution of pH = 1.2 (stimulated gastric solution) at 37 °C, over a period of 8 h, under gentle shaking (50 rpm) using a thermostatic shaker bath (Memmert M00/M01). Very small volumes of the release medium (1 μL) were collected with microsyringes at different intervals of time. The amount of metronidazole was determined spectrophotometrically (Nanodrop ND100, Wilmington, DE, USA) at a wavelength of 277 nm using a calibration curve.

## 3. Results and Discussion

### 3.1. Synthesis of Precursor/Hybrid Microparticles

The synthesis of the hybrid microparticles took place in two steps.

In the first step, precursor microparticles based on glycidyl methacrylate, hydroxyethyl methacrylate and dimethacrylic monomers (EGDMA, DEGDMA and TEGDMA) were synthesized using the suspension polymerization technique. The reaction to obtain the precursor microparticles (AE, AD and AT) is shown in Figure 1.

Table 1 shows the experimental conditions required for the synthesis of precursor microparticles.

From Table 1 it can be seen that suspension polymerization produces high-yield microparticles, regardless of the type of crosslinker used. Similar results were observed for grafting chitosan onto three-dimensional networks based on glycidyl methacrylate and dimethacrylic esters when the grafting reaction occurred directly in the suspension polymerization process [13].

In the second step, hybrid microparticles were synthesized by grafting sodium hyaluronate to the epoxy groups located on the surface of the precursor microparticles; the reaction was carried out in a basic medium under gentle stirring and in a nitrogen atmosphere (Figure 2).

The amount of grafted HA was determined gravimetrically and calculated using the following relationship:(15)$$Q\;\left(\%\right)=\frac{W_{1}-W_{0}}{W_{2}}\cdot 100\;$$
where *W*_0_—the initial amount of microparticles (g); *W*_1_—the amount of grafted microparticles after purification (g); and *W*_2_—the initial amount of HA (g).

#### Optimization of the Grafting Reaction

The optimal conditions for obtaining hybrid microparticles with the highest degree of HA grafting were determined by changing only one reaction parameter (HA concentration, temperature, pH, reaction time) while keeping the other parameters constant. Thus, the influences of the mentioned reaction parameters on the grafting degree of HA are shown in Figure 3 and Figure 4.

From the graphical representations presented above, it can be seen that:The amount of grafted HA increases with the concentration of the polymer solution up to a value of 0.6%, after which equilibrium is reached. This behavior is due to the fact that as the concentration of the HA solution increases, the number of hydroxyl groups that will react in a basic medium with the epoxy groups increases, leading to the formation of a covalent ether -CH_2_-O-HA bond. Additionally, at low concentrations, the viscosity of HA solutions is lower, ensuring a more uniform stirring and better accessibility for the epoxy groups on the surface of the precursor microparticles;Increasing the temperature of the reaction medium has the effect of decreasing the viscosity of the reaction medium and increasing the mobility of the polymer chains, leading to a better interaction between the -OH groups belonging to HA and the epoxy groups on the surface of the precursor microparticles, and thus to a higher amount of grafted HA. Temperature is also known to increase the reaction rate and to favor higher yields for most chemical reactions;Increasing the reaction time to 6 h resulted in an increase in the amount of grafted HA;Another important parameter of the grafting process is the pH of the reaction. In a basic medium, the epoxy ring opening reaction by the -OH group proceeds by an SN2 mechanism and the -OH group is formed at the most substituted atom in the ring. In an acidic environment, the reaction proceeds through the SN1 mechanism, leading to the formation of -OH at the methylene group of the ring, and the rest of the HA molecule, which is huge in volume, encounters significant steric hindrances, making it difficult to bind to the secondary carbon atom of the ring. Thus, the epoxide cycle opening reaction resulting in the grafting of HA to the polymer particles will be increasingly favored by the increasing pH, which intensifies the nucleophilic attack (SN2) of -OH from the polysaccharide to the epoxy ring.

The chemical structure of the crosslinker used to obtain the precursor microparticles also influences the grafting yield. As can be seen, the amount of grafted HA increases in the following order: ADHA < AEHA < ATHA, so the highest amount of grafted HA is recorded in the ATHA hybrid microparticles that were obtained in the presence of TEGDMA as a crosslinker. Thus, increasing the chain length between the two methacrylic groups leads to the formation of microparticles with larger mesh sizes, thus allowing a better interaction between the two reaction partners (precursor microparticles and sodium hyaluronate). A special case is the ADHA microparticles that were obtained in the presence of DEGDMA as a crosslinking agent. In this case, the lower amount of grafted HA is probably due to the more compact structure of the microparticles, which is the result of the complexity of the crosslinking polymerization reaction, a reaction that is often accompanied by internal cyclisation processes.

In conclusion, the most favorable conditions for the synthesis of hybrid AEHA, ADHA and ATHA microparticles are as follows: the concentration of the HA solution = 0.6%, T = 50 °C, t = 6 h, pH = 9, and the ratio of the amount of microparticles: HA = 1:0.6 (g·g^−1^).

### 3.2. Structural Characterization

#### 3.2.1. FTIR Spectroscopy

FTIR spectroscopy was used to highlight HA grafting on precursor microparticles. The infrared spectra of precursor/hybrid microparticles are presented in Figure 5 as well as in Appendix A.

The characteristic absorption bands of AE, AD and AT microparticles are located at: 3452, 3482 and 3515 cm^−1^, characteristic of valence vibrations ν_O-H_ belonging to the HEMA monomer; 2991–2997 cm^−1^ and 2953 cm^−1^ are specific to the symmetric and asymmetric stretching vibrations of the −CH_3_, >CH_2_ and >CH− groups; 1633, 1634 and 1637 cm^−1^ are attributed to the >C=C< bond; 1730 cm^−1^ is characteristic of the >C=O bond of the ester group, which is present in all the chemical structures of the three types of precursor microparticles; 1481 and 1484 cm^−1^ are specific to the bending vibration of the methylene group (δ_CH2_); and 907 cm^−1^ is attributed to the stretching vibrations of the epoxy groups.

The appearance of new absorption bands at 1559, 1539, 1367 and 1341 cm^−1^ indicates the presence in the structures of the hybrid microparticles (AEHA, ADHA and ATHA) of the carboxylate group characteristic of the polysaccharide.

In addition, by comparing the corresponding areas of AE, AD and AT microparticles at wavenumbers 3450, 1152 and 907 cm^−1^ with the specific areas of similar absorption bands for AEHA, ADHA and ATHA microparticles, the following can be observed:The values of the specific absorption band areas at the wavenumber 3450 cm^−1^ are higher for the hybrid microparticles (A_AEHA_ = 46.95 cm^−1^, A_ADHA_ = 28.95 cm^−1^, A_ATHA_ = 78.00 cm^−1^) compared to the values of similar absorption band areas corresponding to the precursor microparticles (A_AE_ = 20.97 cm^−1^, A_AD_ = 13.73 cm^−1^ and A_AT_ = 67.56 cm^−1^). The higher values of the absorption band of the hybrid microparticles are due to the presence of sodium hyaluronate, which has several hydroxyl groups in its structure;The increase in the values of the adsorption band area at 1151 cm^−1^ in the case of the hybrid microparticles is due to the formation of new ether bonds by grafting HA to the epoxy groups from the GMA structure;In the case of the hybrid microparticles, a decrease in the values of the specific areas of the absorption bands is observed from the wavenumber 907 cm^−1^, due to the grafting reaction of HA by the opening of the epoxy ring in a basic medium.

Based on the data obtained from the FTIR spectra, it can be concluded that the grafting reaction of HA on the surface of the precursor microparticles took place successfully.

#### 3.2.2. Dimensional Analysis of Precursor/Hybrid Microparticles

In the case of suspension polymerization, the size and size distribution of the microparticles are influenced by various parameters: the shape of the reaction vessel, the type of stirrer, the stirring speed, the temperature, the chemical structure of the crosslinker or the thermodynamic quality of the porogenic agent used.

The particle size distributions as well as the diameter values of the precursor/hybrid microparticles analyzed using laser diffractometry are shown in Appendix A and Table 2, respectively.

As can be seen from Table 2, the precursor microparticles are micrometric in size, their diameter being influenced by the chemical structure of the crosslinker, i.e., they increase with increasing chain length between the methacrylic groups in the crosslinking agent. Additionally, the hybrid microparticles have larger diameters than the precursor microparticles, leading to the idea that HA has reacted with the epoxy groups to form a layer on the surface covering these microparticles, generating a core-shell structure.

#### 3.2.3. Thermogravimetric Analysis

Thermogravimetric studies were carried out to obtain additional information about the precursor/hybrid microparticles.

Table 3 shows the thermogravimetric characteristics of the precursor/hybrid microparticles, namely: the degradation steps, temperature range for each degradation step, residual mass, activation energy (*E_a_*) and reaction order (*n*).

The thermal behavior of precursor microparticles and sodium hyaluronate is characterized by three stages of thermal decomposition. The first degradation step between 65 and 135 °C (HA), 180–260 °C (AE), 188–222 °C (AD) and 142–154 °C (AT) is characterized by weight losses of 6.20% (HA), 15% (AE), 9.88% (AD) and 6.32% (AT), which are associated in the case of precursor microparticles with the loss of solvents retained in the crosslinked mesh of their structure. The second stage of degradation occurs in the temperature ranges of 225–263 °C (HA), 270–375 °C (AE), 249–349 °C (AD) and 212–308 °C (AT) and is characterized by the highest amount of weight loss: 38.16% (HA), 73.59% (AE), 76.21% (AD) and 38.37% (AT). In this stage, the breakage of the labile bonds occurs first, followed by the destruction of the crosslinked network by the cleavage of the macromolecular chains. The third stage of degradation is in the temperature ranges: 309–510 °C (HA), 380–411 °C (AE), 349–439 °C (AD) and 381–442 °C (AT) with mass losses of 18.21% (HA), 6.46% (AE), 12.34% (AD) and 31.76% (AT).

In the case of the hybrid microparticles, the presence of sodium hyaluronate leads to a slight increase in thermal stability compared to that of the precursor microparticles. The thermal degradation of the microparticles occurs in three steps for AEHA and ADHA microparticles and in four steps for ATHA microparticles. Additionally, as in the case of the precursor microparticles, the second degradation step is characterized by the highest amount of weight loss: 61.27% (AEHA), 67.9% (ADHA) and 28.55% (ATHA). The fourth degradation step specific only to the ATHA microparticles located in the temperature range 361–434 °C is characterized by a weight loss of 19.80%.

The kinetic parameters (activation energy and reaction order) for each thermal decomposition step were determined using the Urbanovici–Segal integral method [22]. If we consider for comparison the second degradation step, which is the step characterized by the highest weight loss, it can be observed that the activation energies for the precursor microparticles have close values (179 kJ·mol^−1^ (AE), 171 kJ·mol^−1^ (AD) and 173 kJ·mol^−1^ (AT)), so the chemical structure of the crosslinker does not influence the way the microparticle degradation takes place. In the case of the hybrid microparticles, however, the activation energies are different (194 kJ·mol^−1^ (AEHA), 174 kJ·mol^−1^ (ADHA), 239 kJ·mol^−1^ (ATHA)) leading to the idea that HA grafting to epoxy groups produces polymeric materials with different chemical structures and thermal stabilities depending on the degree of grafting of the polysaccharide.

#### 3.2.4. Determination of Epoxy Groups

Since hybrid microparticles are obtained by grafting HA to the epoxy groups found in the precursor microparticle structure, it is important to determine their content in the microparticle structure before and after the grafting reaction. The HBr-glacial acetic acid titrimetric method was chosen for the determination of the epoxy groups. The reaction between halogenated acids and the epoxide group results in the opening of the three-atom ring and the formation of a hydroxyl functional group. Table 4 shows the results of the titrimetric method for the determination of the epoxy groups.

From Table 4, it can be seen that the theoretical values obtained for the epoxy groups are higher than those obtained experimentally by titration. By titration, only the epoxide groups can be determined, which are accessible to the HBr reaction, especially those on the surface and those on the layers which are very close to the surface, which partly explains these differences. Another factor to be taken into account is the difference in the chemical composition of the copolymers compared to the starting monomers. For AEHA, ADHA and ATHA microparticles, the number of epoxy groups is reduced due to HA grafting by the ring-opening reaction of the epoxy groups found on the microparticle surface or in the surface layers. However, a small percentage of these epoxy groups remain unreacted, probably due to their reduced accessibility of HBr. From the data in Table 4, it can be seen that grafting was best performed on AT microparticles, the results of which are also in agreement with the amount of grafted HA calculated by the gravimetric method.

### 3.3. Morphological Characterization

#### 3.3.1. Scanning Electron Microscopy

The size, shape and surface morphology of the synthesized polymeric materials were analyzed using scanning electron microscopy. Figure 6 and Figure 7 show micrographs of the precursor/hybrid microparticles, and for easy comparison, SEM images were taken at the same magnification (×5000 for surface structures (small images) and ×500 for microparticle overview (large images)).

The SEM micrographs show that by the chosen synthesis method, spherical particles of micrometric dimensions are obtained and that the surface morphology is influenced by the chemical structure of the crosslinking agent used. Thus, as the alkyl chain between the two methacrylic groups increases, precursor microparticles with a more pronounced porous structure and a rougher surface are obtained. The interaction of the precursor microparticles with sodium hyaluronate results in hybrid microparticles that retain their spherical shape, but the surface morphology changes due to the deposition of a polymer layer on the surface of the precursor microparticles, confirming once again that the grafting reaction of the polysaccharide to the epoxy groups has taken place.

#### 3.3.2. Atomic Force Microscopy

Atomic force microscopy was also used to investigate the surface morphology of the precursor/hybrid microparticles, obtaining information on surface roughness, pore size and geometry. AFM images for AE and AEHA microparticles are shown in Figure 8 as an example.

The AFM images correlate well with those from the scanning electron microscopy, revealing differences in surface morphology between the precursor and hybrid microparticles. Table 5 shows the values of the parameters characteristic of microparticle surfaces.

From the data in Table 5, it can be seen that grafting HA onto precursor microparticles has the effect of decreasing the surface roughness, but also the size of existing pores on the surface. The negative values of S_sk_, a statistical parameter that gives us information about the degree of asymmetry of the distribution of heights on the surface [23], indicate that the two types of microparticles analyzed show porous structures. Additionally, S_ku_ values lower than three confirm that the microparticles present irregular surfaces with various roughness. All these observations reinforce and confirm the conclusions drawn from the SEM analysis. Shape and elongation factors are two important parameters, with which pore structures can be analyzed [24]. The values of these parameters shown in Table 5 indicate that the precursor/hybrid microparticles have elliptical-shaped pores with an irregular outline.

#### 3.3.3. Specific Parameters for Characterizing the Morphology of Porous Structures

The introduction of a pore-forming substance, known as a porogenic agent or diluent, into the organic phase of the reaction system specific to suspension polymerization leads to the formation of permanently heterogeneous structures, i.e., structures containing pores after drying. A highly effective porogenic agent should not react during polymerization but should remain within the microparticle structure until the end of the reaction. When it is removed by extraction, the sites occupied by the porogen become the pores of the crosslinked networks.

Table 6 shows the values of the specific parameters determined to characterize the morphology of the precursor/hybrid microparticles.

From the data presented in Table 6, it can be seen that the pore volume and porosity of the hybrid microparticles increase with the increasing alkyl chain, except for the AD microparticles, whose values decrease. When DEGDMA is used as a crosslinking agent, probably during the crosslinking radical polymerization process, a decrease in the apparent reactivity of the double pendant groups occurs due to steric factors, resulting in the appearance of internal cyclizations and thus the formation of microparticles with more compact structures characterized by lower values of both porosity and pore volume. In the case of the AEHA, ADHA and ATHA microparticles, decreases in pore volume and porosity values compared to those of precursor microparticles are observed, which is due to HA grafting to the epoxy groups on the microparticle surface. Through the grafting reaction, HA coats part of the pores or penetrates into the pores, decreasing their size. This is confirmed by the information obtained by the AFM method as well as by the SEM micrographs. The decrease in pore size was also observed by the AFM method as follows:from 314 to 238 nm for the AE–AEHA microparticle system;from 369 to 280 nm for the AD–ADHA microparticle system;and from 265 to 176 nm for the AT–ATHA microparticle system.

It is also observed that the specific surface area values are higher for hybrid microparticles. This can be explained by the fact that S_sp_ was determined by the dynamic vapor sorption method, and hybrid microparticles due to their hydrophilic structure have a higher capacity to absorb water than precursor microparticles. Additionally, the higher values of the specific surface are due to the fact that the hybrid microparticles have smaller pore sizes than the corresponding precursor microparticles.

The morphology of the pore structures is influenced by the structure and concentration of the monomers, the nature of the porogenic agent and in particular the amount of porogenic agent used. Thus, Figure 9 shows graphical representations of the specific surface area and pore volume values, depending on the amount of porogenic agent used to obtain the AT and ATHA microparticles.

From Figure 9, it can be seen that microparticles with higher porosity structures and more specific surface values are obtained when the amount of porogen agent is increased. For this reason, for the preparation of the precursor microparticles, the dilution was chosen to be 0.6.

### 3.4. Swelling Capacity of Precursor/Hybrid Microparticles in Aqueous Media

Graphical representations of the degree of swelling of precursor/hybrid microparticles versus time in aqueous media with different pH values are shown in Figure 10.

Figure 10 shows that the swelling process is carried out in three stages. In the first stage, the rapid absorption of aqueous solutions of different pH values into the structure of the precursor/hybrid microparticles takes place. The second stage is characterized by a slower absorption, and in the third stage the equilibrium of the swelling process is reached. For AE, AD and AT microparticles, an equilibrium is reached after 960 min and for hybrid microparticles, an equilibrium is reached after 840 min.

The swelling degree of precursor and hybrid microparticles determined with Equation (12) depends on the pore structure and the presence of HA on the microparticle surface. Thus, the degree of swelling for precursor microparticles is not influenced by the pH value of the swelling medium and increases in the order S_W,AD_ < S_W,AE_ < S_w,AT_, similar to the increase in the specific surface values (Table 6). Thus, AT microparticles characterized by a high specific surface area (160 m^2^·g^−1^) show the highest degree of swelling. In the case of the hybrid microparticles, it is observed that the degree of swelling is higher than that corresponding to the precursor microparticles. This is explained by the presence of HA on the surface of the hybrid microparticles, which is a hydrophilic polymer having several -OH groups in its structure. The swelling degrees of the hybrid microparticles in pH = 1.2 are lower than in pH = 5.5. This behavior can be explained by the fact that the COO- groups of the HA molecule in the acidic medium are transformed into COOH groups, thus reducing the electrostatic repulsion between them, and consequently the polymer matrix swells less.

Two mathematical models were used to describe the swelling mechanism in media with different pH values, namely:The second-order kinetic model using the equations [25]:
(16)$$\frac{t}{SR}=\frac{1}{K_{S}\cdot S_{eq}^{2}}+\frac{t}{S_{eq}}\;$$
(17)$$SR\left(g\cdot g^{-1}\right)=\frac{W_{t}-W_{0}}{W_{0}}\;$$
(18)$$S_{eq}\left(g\cdot g^{-1}\right)=\frac{W_{eq}-W_{0}}{W_{0}}$$
where *W_0_*, *W_t_* and *W_eq_* are the amount of precursor/hybrid microparticles at time *t* = 0, *t* = *t* and at equilibrium, respectively [26]. The straight-line plots of *t*/*SR* versus *t* gave the slope of *S_eq_* and intercept *K_S_*.

2.Korsmeyer–Peppas model. The linear form of the Korsmeyer–Peppas equation [27] is given as:

(19)lnF=lnK+n·lnt 
where *F* = *M_t_*/*M**_∞_*, *M_t_*—the amount of water uptake at time t; *M**_∞_*—the amount of water uptake at time approaching infinity; *K*—the swelling rate constant; and *n*—diffusion exponent characteristic for the transport mechanism. The values of *K* and *n* were determined from the linear plots of *lnF* versus *lnt*.

The values of the kinetic parameters obtained by applying the two models are shown in Table 7.

From the data in Table 7, it can be seen that there is a good correlation between the experimental (*S_exp_*) and calculated (*S_eq_*) values and the correlation coefficient *R*^2^ values are greater than 0.997. These results suggest that the swelling mechanism of precursor/hybrid microparticles in aqueous media with different pH values follows the second-order kinetic model. The values of n were in the range 0.141–0.242, indicating that the most likely swelling mechanism is Fickian.

### 3.5. Metronidazole Adsorption and Release Studies

In order to achieve an optimal system with a controlled drug release, the influence of the following parameters must be taken into account: pH; contact time; temperature; and initial drug concentration.

Since metronidazole dissolves in acidic pH, the adsorption process of metronidazole on the precursor/hybrid microparticles was performed from an aqueous solution with pH = 1.2.

Additionally, the effect of contact time is very important for assessing the affinity of precursor/hybrid microparticles for the model drug. Figure 11 shows the influence of contact time on the adsorption capacity of metronidazole on precursor/hybrid microparticles for a metronidazole concentration of 0.5 ×10^−3^ g·mL^−1^ at a temperature of 25 °C.

For precursor microparticles, the contact time for reaching the equilibrium is 720 min, while for hybrid microparticles, the equilibrium is reached at 600 min. Above these contact time values, the amount of drug adsorbed on the precursor/hybrid microparticles remains constant. The shorter time to reach equilibrium indicates a better affinity of the hybrid microparticles for metronidazole compared to that of the precursor microparticles.

Temperature is another important parameter to be taken into account when adsorbing drugs on different polymeric supports, with metronidazole adsorption studies being carried out at 25, 30 and 35 °C (Figure 12a).

Analyzing the graphical representation in Figure 12a, it can be seen that the adsorption of the drug is favored by increasing temperature, an effect that is absolutely expected since the process is of a diffusional nature, causing an increase in the degree of swelling and thus in the diffusion rate of metronidazole into the pores of the precursor/hybrid microparticles.

Increasing the concentration of the drug has the effect of increasing the rate of adsorption. It was also observed that drug adsorption was achieved in higher amount in the case of hybrid microparticles compared to that of precursor microparticles (Figure 12b). This phenomenon is explained by the presence of sodium hyaluronate, which has the role of enhancing the hydrophilicity of the microparticles, leading to a higher degree of swelling and consequently to the adsorption of a higher amount of the drug. In the case of metronidazole adsorption, in an acidic pH mainly physical interactions take place, such as hydrogen bonding between the OH group of metronidazole and the -COOH and -OH groups of the hybrid microparticle structure. The greatest amount of immobilized drug was obtained in the case of the ATHA hybrid microparticles.

In-depth studies of the adsorption process were carried out considering two physico-chemical aspects, namely: the adsorption equilibrium by means of adsorption isotherms, which quantify the interaction between the drug and the support; and adsorption kinetics, which can explain the mechanism of drug adsorption on the solid supports.

#### 3.5.1. Adsorption Equilibrium Studies

For efficient polymer–drug systems, it is important to know how the adsorbate (drug solution) and adsorbent (precursor/hybrid microparticles) interact. For this purpose, the description of the adsorption equilibrium of metronidazole on precursor/hybrid microparticles was performed using the mathematical models of Langmuir, Freundlich, Dubinin–Radushkevich (two-parameter models), Sips and Khan (three-parameter models) isotherms.

The nonlinear forms of the isotherms used can be written as follows:

Two parameter isotherm models:
Langmuir isotherm [28]:
(20)$$\;q_{e}=\frac{q_{m}\cdot K_{L}\cdot C_{e}}{1+K_{L}\cdot C_{e}}\;$$
Freundlich isotherm [29]:
(21)$$\;q_{e\;}=K_{F}\cdot C_{e}^{\frac{1}{n_{f}}}\;$$
Dubinin–Radushkevich isotherm [30]:
(22)$$\;q_{e}=q_{m\;}exp\left(-K_{D}\cdot \epsilon_{D}^{2}\right)\;$$
where *q_e_* is the metronidazole amount adsorbed at equilibrium (mg·g^−1^); *q_m_* is the maximum adsorption capacity (mg·g^−1^); *K_L_* is the Langmuir constant that reflects the affinity between the adsorbate and the adsorbent (L·g^−1^); *K_F_* is the adsorption capacity for a unit’s equilibrium concentration (L·g^−1^); 1/n_F_ is a constant that suggests the favorability and capacity of the adsorbent–adsorbate system; ε is the Polanyi potential; and *K_D_* is the constant which is related to the calculated average sorption energy *E* (kJ·mol^−1^).The constant *K_D_* can give the valuable information regarding the mean energy of adsorption by the equation:
(23)$$E={\left(-2K_{D}\right)}^{1/2}$$



2.Three-parameter isotherm models:
Sips isotherm [31]:
(24)$$q_{e}=\frac{q_{m}\cdot K_{S}\cdot C_{e}^{n_{S}}}{1+K_{S}\cdot C_{e}^{n_{S}}}$$
Khan isotherm [32]:
(25)$$\;q_{e}=q_{m}\frac{b_{K}\cdot C_{e}}{{\left(1+b_{K}\cdot C_{e}\right)}^{n_{K}}}$$
where *K_S_* is the Sips constant (L·mg^−1^); *n_S_* is the Sips model exponent; *b_K_* is the Khan model constant; and *n_K_* is the Khan model exponent.


Sips and Khan isotherms represent the combined features of the Langmuir and Freundlich isotherm equations. Thus, in the case of a low adsorbent concentration, the Sips isotherm is reduced to the Freundlich isotherm, while at high adsorbate concentrations, it shows the characteristics of the Langmuir isotherm [33] Additionally, if *n_K_* = 1, Equation (24) can be simplified to the Langmuir isotherm equation, whereas if *n_K_·C_e_* >> 1, Equation (25) can be approximated by the Freundlich type isotherm equation [32].

The isotherm model plots of metronidazole adsorption onto precursor/hybrid microparticles are illustrated in Figure 13, while the parameters and the statistical error functions values (*R*^2^ and χ^2^) are presented in Table 8 and Table 9.

From the analysis of the data presented in Table 8 and Table 9, it can be seen that:The values of the maximum adsorption capacity, *q_m_*, calculated based on the Langmuir, Dubinin–Radushkevich, Sips and Khan models, are close to the experimental values;With increasing temperatures, the saturation capacity increases, indicating a better accessibility to the adsorption centers on the surface of the precursor/hybrid microparticles;The Langmuir constant values increase with increasing temperatures, thus showing a higher metronidazole adsorption efficiency at higher temperatures;The highest value of *K_L_* was obtained for the ATHA microparticles;The values of the constant 1/n_F_ are in the range between 0 and 1, which would indicate that the Freundlich isotherm is favorable for metronidazole adsorption on precursor/hybrid microparticles;The values of *E* are in the range of 3.47–6.97 kJ/mol, indicating that the adsorption process of metronidazole on both the precursor and hybrid microparticles is of a physical nature;The values of the exponents *n_S_* and *n_K_* are very close to unity, which provides a further argument that the adsorption process of metronidazole on precursor/hybrid microparticles is better suited to the Langmuir model than to the Freundlich model;The value of the Khan constant, *b_K_*, increases with increasing temperatures and has the highest values when using the ATHA hybrid microparticles;The values close to unity for the correlation coefficient *R*^2^ that are associated with low values of the χ^2^ test indicate that the Langmuir, Dubinin–Radushkevich, Sips and Khan isotherms apply quite well to the experimental data obtained for metronidazole adsorption on precursor/hybrid microparticles;Lower values of *R*^2^ and higher values of χ^2^ obtained from the application of the Freundlich isotherm indicate that this isotherm does not describe the experimental data well.

#### 3.5.2. Kinetic Studies

In order to investigate the mechanism of metronidazole adsorption on precursor/hybrid microparticles, the experimental data were interpreted using four mathematical models, namely: the Lagergren model (pseudo-first order kinetic model), the Ho model (pseudo-second order kinetic model), the Elovich model and the Weber–Morris intraparticle diffusion model. The nonlinear forms of the Lagergren (Equation (26)) [34], Ho (Equation (27)) [35] and Elovich models (Equation (28)) [36], as well as the linear form of the Weber–Morris model (Equation (29)) [37] are written below:(26)$$\;q_{t}=q_{e}\left(1-e^{-k_{1}t}\right)$$
(27)$$\;q_{t}=\frac{k_{2}\cdot q_{e}^{2}\cdot t}{1+k_{2}\cdot q_{e}\cdot t}$$
(28)$$\;q_{t}=\frac{1}{\beta }\ln\left(1+\alpha \cdot \beta \cdot t\right)$$
(29)$$\;q_{t}=k_{id}\cdot t^{0.5}+C_{i}$$
where k_1_ is the rate constant of the pseudo-first order model (min^−1^); k_2_ is the rate constant of the pseudo-second order model (g·mg^−1^·min^−1^); α is the initial adsorption rate (mg·g^−1^· min^−1^); β is the desorption constant (g·mg^−1^); k_id_ is the intraparticle diffusion rate constant (g·mg^−1^·min^−0.5^); and *C_i_* is the constant that gives information about the thickness of the boundary layer.

Figure 14a,b presents the nonlinear plots of the Lagergren, Ho and Elovich models as well as the straight-line plots of the Weber–Morris model in case of metronidazole adsorption (C_metronidazole_ = 1 × 10^−3^ g·mL^−1^) on the precursor/hybrid microparticles at 35 °C.

The kinetic parameters obtained from the Lagergren, Ho, Elovich and Weber–Morris models are presented in Table 10 and Table 11.

From the data presented in Table 10 and Table 11, it can be seen that the calculated adsorption capacity values based on the first-order kinetic model are very close to the experimental values for metronidazole adsorption on the precursor/hybrid microparticles. The values of the rate constant k_1_ increase with increasing temperatures, indicating a higher adsorption rate of the drug at higher temperatures. It is also observed that values of R^2^ are very close to unity and are associated with low values of χ^2^, showing that the first-order kinetic model describes the experimental data quite well. These results suggest that the adsorption of metronidazole on precursor/hybrid microparticles is of a physical nature. By applying the second-order kinetic model, it can be seen that the values of q_e,calc_ are not as close to the values of q_e,exp_ as obtained in the case of applying the first-order kinetic model. The relatively high values of R^2^ associated with the high values of χ^2^ indicate that the second-order kinetic model does not describe the experimental data well in the case of metronidazole adsorption on the precursor/hybrid microparticles. Additionally, the value of the rate constant k_2_ increases with increasing temperatures, again asserting that the rate of drug adsorption is higher at higher temperatures. The lower values of R^2^ were correlated with higher values of χ^2^ obtained for metronidazole adsorption on precursor/hybrid microparticles; hence, the application of the Elovich model provides a further argument that the metronidazole adsorption is not chemical in nature.

Additionally, from Table 10 and Table 11, it can be seen that the C_i2_ values increase with increasing temperatures, thus indicating increasing boundary layer thickness associated with decreasing external mass transfer and increasing internal mass transfer. The highest values of C_i2_ were obtained for hybrid microparticles, confirming that they are good adsorbents. The results obtained by applying the Weber–Morris model lead us to the conclusion that intraparticle diffusion is not the only process influencing the adsorption rate.

#### 3.5.3. Thermodynamic Studies

The adsorbent performance of the precursor/hybrid microparticles was also demonstrated by thermodynamic studies. For this purpose, the following thermodynamic parameters were calculated: Gibbs free energy changes (Δ*G*), enthalpy change (Δ*H*) and entropy change (Δ*S*).

The values of Δ*H* and Δ*S* were estimated using Van’t Hoff equation [38]:(30)$$\ln K=\frac{\Delta S}{R}-\frac{\Delta H}{RT}$$
where *K*—the Langmuir adsorption equilibrium constant obtained at different temperature values [39]; *R*—the ideal gas constant (8.314 J·mol^−1^·K^−1^); and *T*—temperature in Kelvin.

The Δ*G* value was calculated using the thermodynamic equation:(31)ΔG=ΔH−T·ΔS 

According to the data in the literature, the values of Δ*H* and Δ*G* can provide information about the type of adsorption process [40]. The linear plot of lnK versus 1/*T* gives us the thermodynamic parameters of the adsorption process of metronidazole on precursor/hybrid microparticles, and their values are shown in Table 12.

From the data presented in Table 12, it can be seen that:∆*H* values < 40 kJ·mol^−1^ indicate that the interactions between precursor/hybrid microparticles and metronidazole are physical in nature.The positive enthalpy value, ∆H, demonstrates that the adsorption process studied is endothermic.The negative values of ∆G indicate that the adsorption processes of metronidazole on the precursor/hybrid microparticles are spontaneous and as the temperatures increase, the negative value of the parameter increases in absolute value, which demonstrates that the adsorption of the drug is favorable at higher temperatures.The positive values of entropy, ΔS, suggest the affinity of precursor/hybrid microparticles for metronidazole, and this affinity increases with increasing temperatures.

#### 3.5.4. Release Studies

The goal of the research was to obtain a microparticulate system capable of the controlled/sustained release of the adsorbed drug. For this reason, after loading the precursor/hybrid microparticles with metronidazole, kinetic release studies were performed. Release studies have been conducted for the precursor–drug microparticle and respectively for the hybrid–drug microparticle systems containing the highest amount of the included drug. The release profiles are represented in Figure 15.

From the graphical representations, it can be seen that the release process of metronidazole from the precursor microparticles occurs at a higher rate than that for the hybrid microparticles.

The interpretation of the metronidazole release kinetics from precursor/hybrid microparticle–drug systems was performed using three mathematical models:
Higuchi model [41]:
(32)$$\;Q_{t}=k_{H}\cdot t^{0.5}$$
Korsmeyer–Peppas model [27]:
(33)$$\frac{M_{t}}{M_{\infty }}=k_{r}\cdot t^{n}$$
Baker–Lansdale model [42]:
(34)$$\frac{3}{2}\left[1-{\left(1-\frac{M_{t}}{M_{\infty }}\right)}^{\frac{2}{3}}\right]-\frac{M_{t}}{M_{\infty }}=k_{BL}\cdot t\;$$
where *Q_t_*—the amount of drug released at time t; *k_H_*—the Higuchi dissolution constant; *M_t_*/*M_∝_*—the fraction of drug released at time t; *k_r_*—the release rate constant that is characteristic for drug–polymeric interactions; *n*—the diffusion exponent that is characteristic for the release mechanism; and *k_BL_*—the release constant.

The values of the release parameters are presented in Table 13.

The rate constants obtained by applying the three kinetic models indicate that the release rate of metronidazole from the precursor microparticles is higher than that for the hybrid microparticles. The different amounts of drug released can be explained by the physical interactions of metronidazole with the functional groups belonging to the chemical structure of the hybrid microparticles. In the case of precursor microparticles, the drug is retained in a larger quantity on the surface and for this reason can be released at a higher rate.

The value of the diffusion exponent n calculated on the basis of the Korsmeyer–Peppas model further argues that:In the case of precursor microparticles, the value of n < 0.43 indicates that the release mechanism of metronidazole is a Fick-type diffusion mechanism;In the case of hybrid microparticles, the value of n is in the range of 0.57–0.63, indicating that the release mechanism of metronidazole is a complex mechanism, controlled by both diffusion and swelling processes characteristic of an anomalous or non-Fickian diffusion;The values of the n parameter are less than 0.85, leading to the conclusion that the microparticles swelled but did not undergo any disintegration or erosion.

Similar results have been found in the literature for other microparticulate systems. For example, in case of microparticles based on gelatin and poly(ethylene glycol) coated with ethyl cellulose, the metronidazole release rates and transport parameters have suggested the non-Fickian mechanism [43]. Additionally, the release kinetics of the metronidazole from the hydrogel containing crosslinked chitosan microparticles best fit the Higuchi model [44].

## 4. Conclusions

By aqueous suspension polymerisation, three series of porous microparticles were obtained based on GMA, HEMA and one of the following crosslinking agents: EGDMA, DGDMA and TEGDMA. By the grafting reaction of sodium hyaluronate to the existent epoxy groups on the surface of AE, AD and AT microparticles, hybrid porous microparticles were obtained.

Precursor/hybrid microparticles were structurally characterized by appropriate techniques: FTIR spectroscopy, epoxy groups content, thermogravimetric analysis, dimensional analysis, and grafting degree of HA. From a morphological point of view the precursor/hybrid microparticles were characterized by: scanning electron microscopy, atomic force microscopy, and specific parameters for the characterization of the morphology of porous structures.

The information from the data acquired using the above-mentioned techniques showed that spherical microparticles of micrometer size with different surface morphologies depending on the synthesis conditions were obtained by suspension polymerization, and the grafting reaction of HA on the surface of the precursor microparticles in a basic medium was a success.

The swelling ability of precursor/hybrid microparticles in aqueous media with different pH values was studied, and the mechanism by which the swelling of precursor/hybrid microparticles in aqueous solutions with different pH values occurred is Fick-type and follows the second-order kinetic model.

The adsorptive performance of the precursor/hybrid microparticles has been shown by kinetic, thermodynamic and equilibrium studies. The experimental data obtained in the case of the metronidazole adsorption on precursor/hybrid microparticles were described using the nonlinear forms of Langmuir, Freundlich, Dubinin–Radushkevich, Sips and Khan isotherms. Adsorption isotherms demonstrate that the adsorption of metronidazole on precursor/hybrid microparticles occurs according to a monolayer adsorption.

To explain the mechanism of metronidazole adsorption on precursor/hybrid microparticles, the experimental data were modelled using four kinetic models, namely: first-order kinetic model, second-order kinetic model, Elovich model and Weber–Morris intraparticle diffusion model. The first-order kinetic model describes the experimental data quite well for metronidazole adsorption on both precursor and hybrid microparticles.

The release kinetics reflect that the release mechanism of metronidazole is a Fick-type diffusion mechanism in the case of precursor microparticles, while in the case of hybrid microparticles, it is a complex mechanism characteristic of anomalous or non-Fickian diffusion.

## Figures and Tables

**Figure 1 polymers-14-04151-f001:**
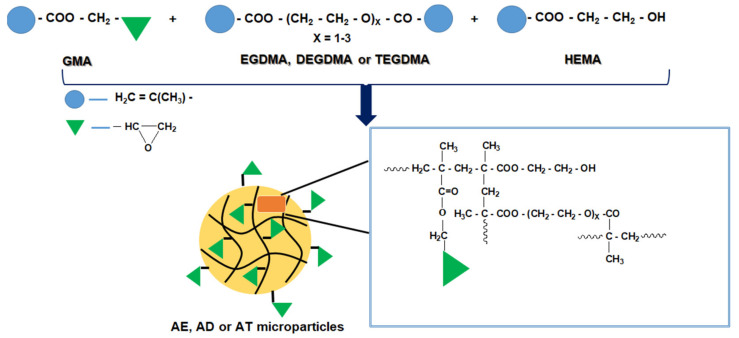
Schematic representation of the reaction to obtain precursor microparticles (AE, AD or AT).

**Figure 2 polymers-14-04151-f002:**
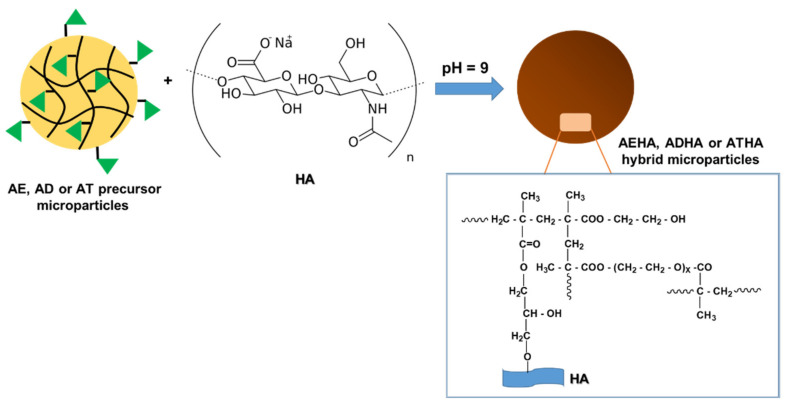
Schematic representation of the reaction to obtain hybrid microparticles (AEHA, ADHA or ATHA).

**Figure 3 polymers-14-04151-f003:**
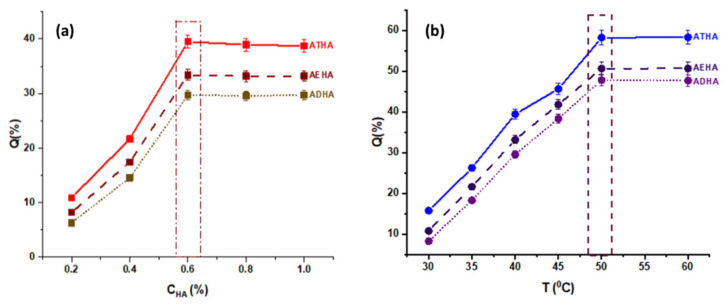
(**a**) The influence of HA concentration on amount of grafted polymer (T = 35 °C, t = 6 h, pH = 9); and (**b**) the influence of temperature on amount of grafted polymer (C_P_ = 0.6%, t = 6 h, pH = 9).

**Figure 4 polymers-14-04151-f004:**
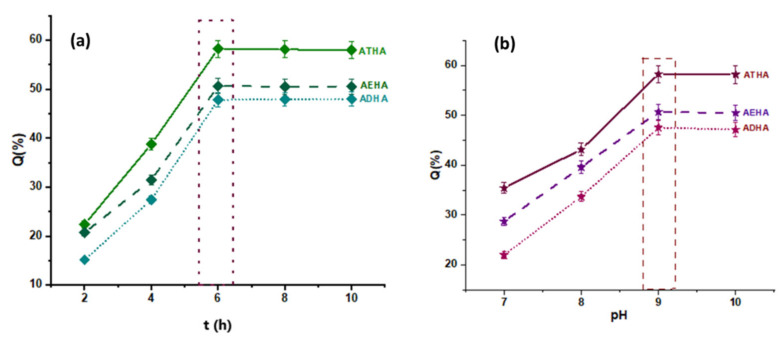
(**a**) The influence of the reaction time on the amount of grafted polymer (C_P_ = 0.6%, T = 50 °C, pH = 9); and (**b**) the influence of the pH of the HA solution on amount of grafted polymer (C_P_ = 0.6%, T = 50 °C, t = 6 h).

**Figure 5 polymers-14-04151-f005:**
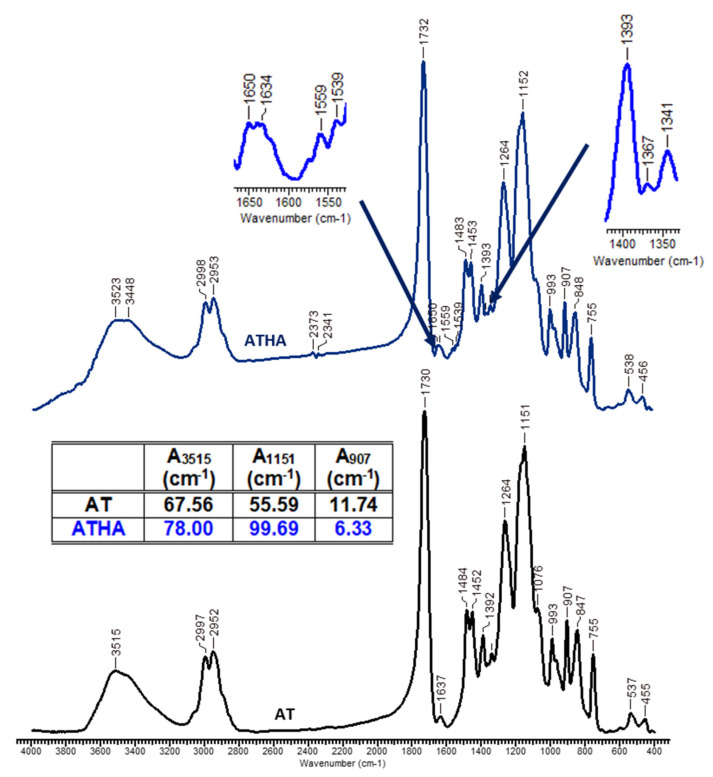
The infrared spectra of AT and ATHA microparticles.

**Figure 6 polymers-14-04151-f006:**
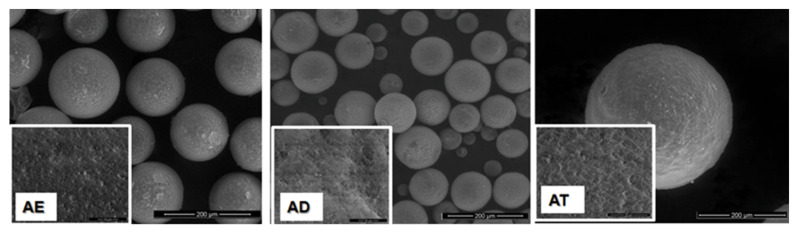
SEM micrographs of precursor microparticles.

**Figure 7 polymers-14-04151-f007:**
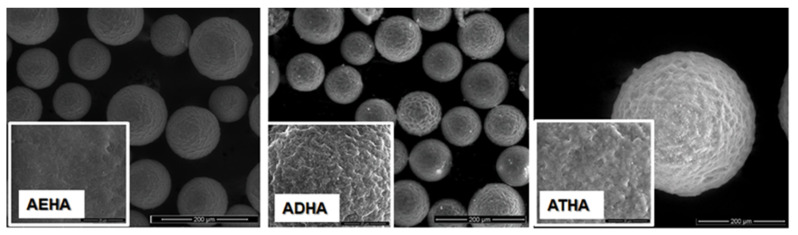
SEM micrographs of hybrid microparticles.

**Figure 8 polymers-14-04151-f008:**
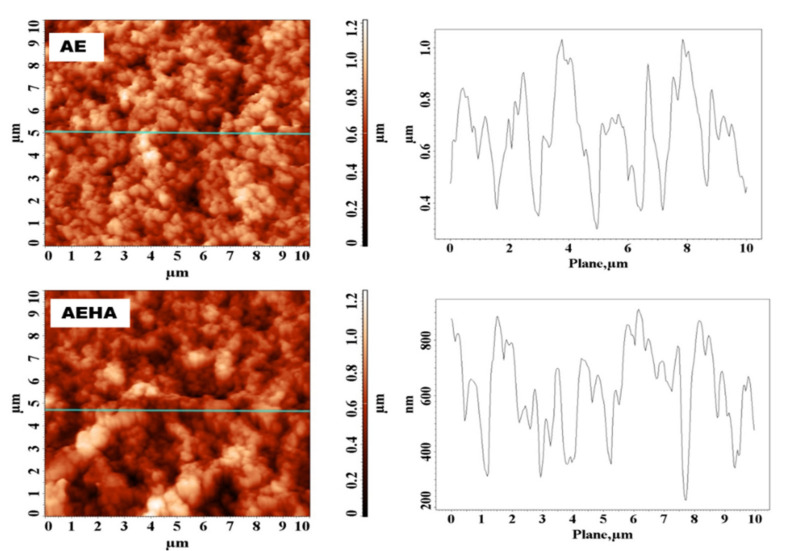
AFM images (2D) and cross-section profile of AE and AEHA microparticles.

**Figure 9 polymers-14-04151-f009:**
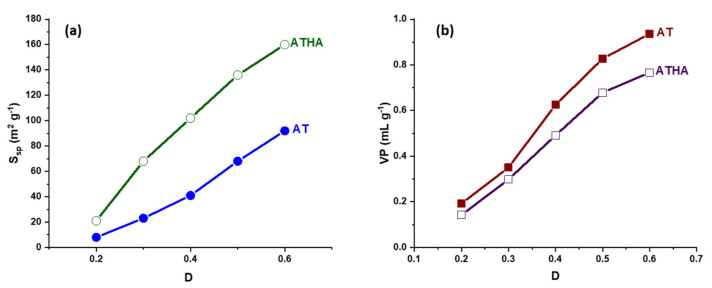
The influence of the amount of porogenic agent on the specific surface area (**a**) and the pore volume (**b**) for AT and ATHA microparticles.

**Figure 10 polymers-14-04151-f010:**
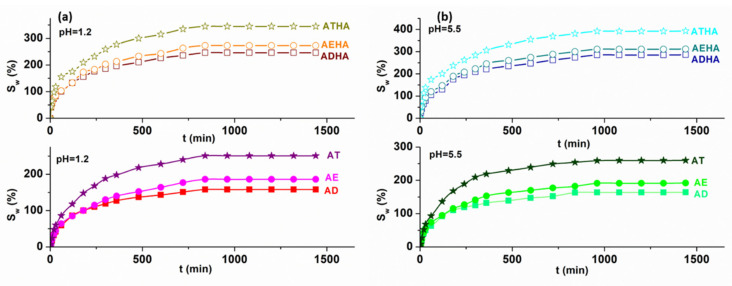
Time dependence of the degree of swelling in pH = 1.2 (**a**) and pH = 5.5 (**b**) for precursor/hybrid microparticles.

**Figure 11 polymers-14-04151-f011:**
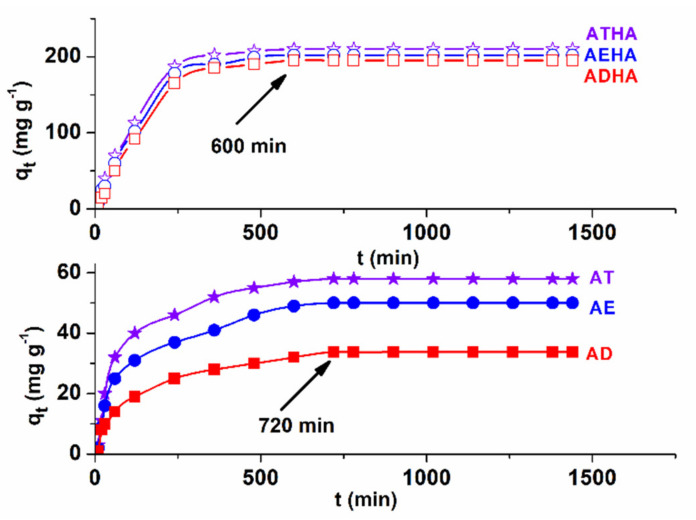
Influence of contact time on the adsorption capacity of metronidazole on precursor/hybrid microparticles.

**Figure 12 polymers-14-04151-f012:**
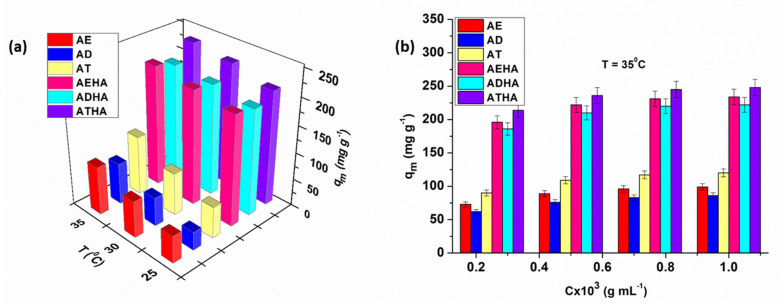
Influence of temperature (**a**) and the initial concentration of metronidazole (**b**) on the adsorption capacity of the drug onto precursor/hybrid microparticles.

**Figure 13 polymers-14-04151-f013:**
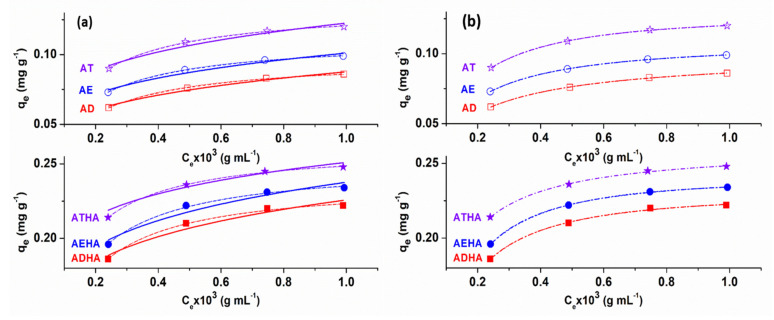
The results of nonlinear fits of Langmuir (dash line); Freundlich (solid line); Dubinin–Radushkevich (short dot line) (**a**); Sips (dash dot line); and Khan (short dash line). (**b**) Isotherm for metronidazole adsorption on precursor/hybrid microparticles at T = 35 °C.

**Figure 14 polymers-14-04151-f014:**
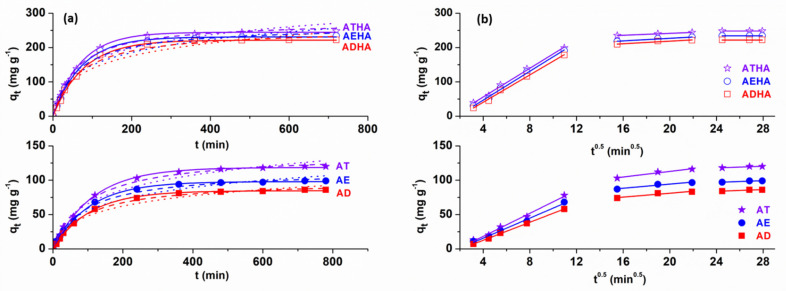
Lagergren (solid line), Ho (dash line), Elovich (dot line) models (**a**) and Weber–Morris intraparticle diffusion model (**b**) for metronidazole adsorption on precursor/hybrid microparticles (C_metronidazole_ = 1 × 10^−3^ g·mL^−1^, T = 35 °C).

**Figure 15 polymers-14-04151-f015:**
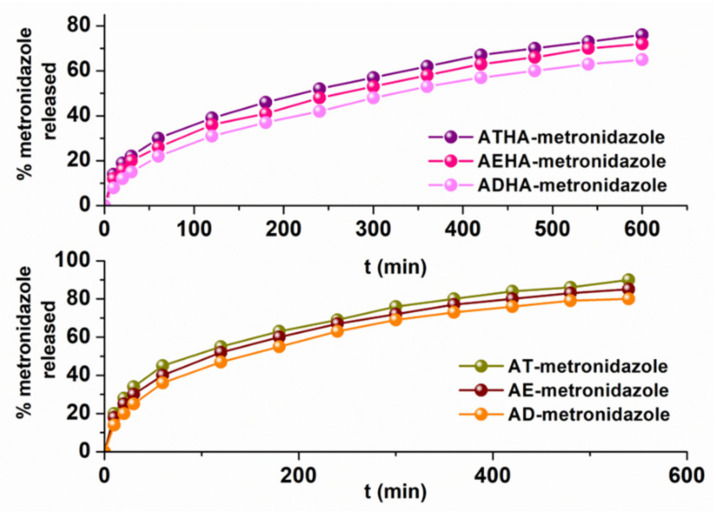
Metronidazole release profiles from precursor/hybrid microparticles (pH = 1.2).

**Table 1 polymers-14-04151-t001:** The experimental conditions for the synthesis of precursor microparticles.

Sample Code	GMA (% mol)	HEMA (% mol)	EGDMA (% mol)	DEGDMA (% mol)	TEGDMA (% mol)	Porogenic Agent	Dilution	Reaction Yield (%)
AE	70	20	10	-	-	butyl acetate	0.6	93
AD	70	20	-	10	-	butyl acetate	0.6	90
AT	70	20	-	-	10	butyl acetate	0.6	96

**Table 2 polymers-14-04151-t002:** Diameters of precursor/hybrid microparticles.

	AE	AEHA	AD	ADHA	AT	ATHA
D_m_ (μm)	124	135	165	173	184	194

**Table 3 polymers-14-04151-t003:** Thermogravimetric characteristics of precursor/hybrid microparticles.

Sample Code	Decomposition Temperature			Residual Mass (%)	*E_a_* (kJ·mol^−1^)	*n*	*R^2^*
	*T_i_* (°C)	*T_m_* (°C)	*T_f_* (°C)				
AE	180	240	260	5.181	121	1.7	0.992
	270	332	375		179	1.8	0.992
	380	411	430		327	1.3	0.998
AD	188	209	222	1.57	149	1.7	0.996
	249	321	349		171	1.8	0.994
	349	417	439		223	1.8	0.994
AT	142	150	154	22.88	114	1.9	0.996
	212	242	308		173	1.8	0.992
	381	410	442		192	1.7	0.993
AEHA	180	231	240	14.68	133	1.4	0.991
	280	341	360		194	1.7	0.992
	380	416	460		425	2.6	0.994
ADHA	185	199	252	1.38	159	1.7	0.993
	252	291	342		174	1.9	0.994
	342	411	432		256	1.7	0.994
ATHA	72	94	115	36.30	87	1.7	0.997
	213	253	273		239	1.8	0.996
	292	299	321		323	1.8	0.997
	361	407	434		476	1.8	0.996
HA	65	102	135	34.82	62	1.4	0.993
	225	263	309		127	1.7	0.997
	309	411	510		310	1.9	0.997

**Table 4 polymers-14-04151-t004:** The values of epoxy groups obtained theoretically and experimentally by titration.

Sample Code	Epoxy Groups			
	Theoretical		Experimental	
	mmol·g^−1^	%	mmol·g^−1^	%
AE	4.80	20.71	2.82	12.17
AD	4.67	20.10	2.55	10.98
AT	4.54	19.52	3.22	13.86
AEHA			1.01	4.32
ADHA			1.56	6.71
ATHA			0.30	1.30

**Table 5 polymers-14-04151-t005:** Surface profile parameters for precursor/hybrid microparticles.

Sample Code	S_a_ (nm)	S_q_ (nm)	S_sk_	S_ku_	d_med_ (nm)	f_shape_	f_elongation_
AE	124.2	131.3	−0.686	0.310	314	0.363	0.485
AD	143.3	156.1	−0.423	0.486	369	0.391	0.426
AT	156.8	172.5	−0.143	0.532	265	0.426	0.375
AEHA	95.4	110.2	−0.514	0.195	238	0.174	0.228
ADHA	112.9	129.9	−0.365	0.371	280	0.248	0.269
ATHA	126.6	143.7	−0.089	0.498	176	0.323	0.280

**Table 6 polymers-14-04151-t006:** Porosity parameters of precursor/hybrid microparticles.

Sample Code	VP (mL·g^−1^)	P (%)	S_sp_ (m^2^·g^−1^)
AE	0.7073	45	78
AD	0.5076	37	54
AT	0.9363	52	92
AEHA	0.6004	41	120
ADHA	0.4461	34	86
ATHA	0.7668	47	160

**Table 7 polymers-14-04151-t007:** Kinetic parameters of the swelling process in various aqueous solutions.

Sample Code	Second-Order Model				Korsmeyer–Peppas Model		
	*S_exp_* (g·g^−1^)	*K_S_* (g·g^−1^)	*S_eq_* (g·g^−1^)	*R^2^*	*K*	*n*	*R^2^*
pH = 1.2							
AE	1.86	0.0033	2.07	0.997	0.282	0.172	0.999
AD	1.58	0.0026	1.73	0.998	0.212	0.164	0.998
AT	2.51	0.0045	2.80	0.997	0.315	0.225	0.998
AEHA	2.73	0.0035	2.93	0.997	0.302	0.145	0.998
ADHA	2.46	0.0030	2.61	0.997	0.216	0.141	0.999
ATHA	3.45	0.0050	3.69	0.997	0.325	0.239	0.999
pH = 5.5							
AE	1.92	0.0028	2.09	0.997	0.234	0.200	0.997
AD	1.64	0.0025	1.77	0.997	0.205	0.195	0.999
AT	2.60	0.0030	2.84	0.998	0.266	0.237	0.998
AEHA	3.12	0.0039	3.35	0.997	0.248	0.188	0.997
ADHA	3.08	0.0030	2.86	0.997	0.209	0.174	0.997
ATHA	3.93	0.0054	4.19	0.998	0.294	0.242	0.998

**Table 8 polymers-14-04151-t008:** Two- and three-parameter isotherm values for adsorption of metronidazole onto AE, AD and AT microparticles.

Isotherm Model	Parameter	AE			AD			AT		
		25 °C	30 °C	35 °C	25 °C	30 °C	35 °C	25 °C	30 °C	35 °C
Langmuir	q_m_ (mg·g^−1^)	72.93	94.66	112.25	55.56	66.23	92.22	84.89	103.02	128.43
	K_L_ (L·g^−1^)	5.24	5.98	6.79	4.07	4.64	5.17	6.31	7.63	9.24
	R^2^	0.997	0.996	0.998	0.999	0.997	0.996	0.995	0.998	0.999
	χ^2^	0.143	0.566	0.163	0.025	0.218	0.356	0.891	0.360	0.161
Freundlich	K_F_ (L·g^−1^)	60.70	80.25	101.08	43.16	62.32	87.65	70.28	95.68	122.57
	1/n_F_	0.32	0.29	0.21	0.38	0.28	0.23	0.32	0.31	0.20
	R^2^	0.949	0.947	0.952	0.972	0.969	0.968	0.941	0.956	0.941
	χ^2^	52.148	83.909	64.88	14.212	25.898	36.104	79.554	101.484	107.284
Dubinin-Radushkevich	q_m_ (mg·g^−1^)	65.39	85.82	106.01	47.16	66.23	92.28	75.73	103.02	128.43
	E (kJ·mol^−1^)	3.80	4.06	4.86	3.47	3.91	4.67	3.86	4.12	4.97
	R^2^	0.999	0.999	0.999	0.996	0.997	0.997	0.998	0.998	0.999
	χ^2^	0.293	0.285	0.252	0.257	0.186	0.162	0.335	0.064	0.061
Sips	q_m_ (mg·g^−1^)	67.14	87.98	108.98	55.18	72.91	100.08	75.82	108.28	129.30
	K_S_ (L·mg^−1^)	5.31	6.03	7.17	4.25	4.88	5.25	7.30	8.94	10.30
	n_S_	1.15	1.13	1.13	1.01	0.95	0.98	1.15	1.16	1.14
	R^2^	0.999	0.998	0.998	0.999	0.999	0.997	0.999	0.997	0.998
	χ^2^	0.393	0.209	0.094	0.214	0.275	0.239	0.250	0.619	0.154
Khan	q_m_ (mg·g^−1^)	85.51	100.91	120.95	57.11	71.42	95.51	94.88	113.90	144.25
	b_K_	2.11	3.17	4.38	1.05	2.46	3.41	2.66	3.03	5.88
	n_K_	1.09	1.01	1.04	1.01	0.99	0.99	1.10	1.10	1.08
	R^2^	0.998	0.999	0.998	0.998	0.999	0.997	0.999	0.998	0.997
	χ^2^	0.143	0.097	0.168	0.177	0.043	0.125	0.037	0.142	0.166

**Table 9 polymers-14-04151-t009:** Two- and three-parameter isotherm values for adsorption of metronidazole onto AEHA, ADHA and ATHA microparticles.

Isotherm Model	Parameter	AEHA			ADHA			ATHA		
		25 °C	30 °C	35 °C	25 °C	30 °C	35 °C	25 °C	30 °C	35 °C
Langmuir	q_m_ (mg·g^−1^)	227.75	237.60	250.99	221.75	230.22	238.39	233.79	251.62	262.05
	K_L_ (L·g^−1^)	21.20	24.71	29.17	12.98	14.68	16.74	23.08	28.32	34.58
	R^2^	0.995	0.996	0.998	0.995	0.996	0.998	0.997	0.998	0.998
	χ^2^	0.721	0.579	0.369	0.608	0.256	0.149	0.142	0.134	0.156
Freundlich	K_F_ (L·g^−1^)	216.32	225.40	237.72	209.33	218.16	225.80	223.80	240.63	251.10
	1/n_F_	0.13	0.12	0.12	0.13	0.13	0.12	0.11	0.10	0.10
	R^2^	0.934	0.927	0.923	0.923	0.929	0.930	0.950	0.920	0.947
	χ^2^	153.564	181.197	228.701	180.312	166.542	190.317	92.568	171.073	123.686
Dubinin-Radushkevich	q_m_ (mg·g^−1^)	282.36	300.36	312.01	256.49	279.21	295.56	296.41	311.17	320.38
	E (kJ·mol^−1^)	6.24	6.31	6.38	6.00	6.27	6.33	6.71	6.77	6.97
	R^2^	0.999	0.998	0.997	0.998	0.997	0.998	0.999	0.997	0.998
	χ^2^	0.087	0.603	0.207	0.066	0.289	0.125	0.212	0.209	0.057
Sips	q_m_ (mg·g^−1^)	222.61	230.17	242.42	213.49	223.37	232.16	233.96	244.28	260.19
	K_S_ (L·mg^−1^)	22.67	26.47	30.10	12.09	14.23	17.39	23.75	29.90	35.34
	n_S_	1.14	1.22	1.15	1.17	1.19	1.14	0.99	1.06	1.07
	R^2^	0.997	0.999	0.998	0.998	0.999	0.997	0.997	0.997	0.997
	χ^2^	0.348	0.130	0.267	0.072	0.182	0.391	0.254	0.186	0.172
Khan	q_m_ (mg·g^−1^)	233.19	240.25	254.73	227.07	232.27	241.19	241.44	257.32	269.98
	b_K_	2.53	3.86	4.92	1.76	2.90	3.84	3.79	4.52	6.95
	n_K_	1.03	1.04	1.05	1.06	1.04	1.04	1.00	1.04	1.01
	R^2^	0.997	0.999	0.998	0.998	0.999	0.997	0.998	0.997	0.997
	χ^2^	0.323	0.136	0.191	0.137	0.074	0.185	0.126	0.153	0.167

**Table 10 polymers-14-04151-t010:** Kinetic parameters for adsorption of metronidazole onto AE, AD and AT microparticles.

	AE			AD			AT		
	25 °C	30 °C	35 °C	25 °C	30 °C	35 °C	25 °C	30 °C	35 °C
q_e,exp_ (mg·g^−1^)	59.00	78.00	99.00	42.00	61.00	86.00	68.00	93.00	120.00
*Lagergren model*									
q_e,calc_ (mg·g^−1^)	57.97	76.99	97.92	41.78	60.44	84.60	67.28	91.68	118.53
k_1_ × 10^2^ (min^−1^)	0.953	0.989	1.00	0.80	0.89	0.95	1.01	1.13	1.23
R^2^	0.998	0.998	0.998	0.997	0.998	0.998	0.998	0.999	0.999
χ^2^	1.228	1.317	1.257	1.248	1.209	1.228	0.656	0.311	0.431
*Ho model*									
q_e,calc_ (mg·g^−1^)	66.53	89.09	113.94	48.65	72.01	98.64	76.47	106.10	139.07
k_2_ × 10^5^ (g·mg^−1^·min^−1^)	10.88	12.01	22.68	9.54	10.84	19.51	7.16	13.01	18.92
R^2^	0.991	0.995	0.994	0.989	0.996	0.996	0.987	0.995	0.995
χ^2^	4.629	4.128	4.554	9.230	4.201	6.215	9.396	6.081	9.206
*Elovich model*									
α (mg·g^−1^·min^−1^)	1.37	1.62	1.93	0.80	0.89	1.62	1.83	1.93	2.11
β (g·mg^−1^)	0.06	0.04	0.03	0.08	0.05	0.04	0.06	0.04	0.03
R^2^	0.964	0.975	0.974	0.963	0.981	0.977	0.953	0.975	0.979
χ^2^	59.661	24.428	40.029	50.971	11.864	27.106	34.217	33.678	46.254
*Weber–Morris intraparticle diffusion model*									
k_id_ (mg·g^−1^·min-^0.5^)	1.10	1.18	1.27	0.44	0.47	0.54	1.42	1.49	1.65
C_i2_	25.10	36.30	44.37	22.62	33.21	40.90	29.03	39.94	48.96
R^2^	0.994	0.995	0.996	0.993	0.994	0.994	0.995	0.995	0.996

**Table 11 polymers-14-04151-t011:** Kinetic parameters for adsorption of metronidazole onto AEHA, ADHA and ATHA microparticles.

	AEHA			ADHA			ATHA		
	25 °C	30 °C	35 °C	25 °C	30 °C	35 °C	25 °C	30 °C	35 °C
q_e,exp_ (mg·g^−1^)	213.00	222.00	234.00	206.00	215.00	221.00	221.00	237.00	248.00
*Lagergren model*									
q_e,calc_ (mg·g^−1^)	212.67	221.74	230.96	207.12	213.36	220.58	218.68	234.06	244.72
k_1_ × 10^2^ (min^−1^)	1.16	1.28	1.34	0.93	0.96	1.10	1.35	1.38	1.43
R^2^	0.999	0.998	0.998	0.998	0.998	0.998	0.997	0.998	0.998
χ^2^	1.057	1.287	1.268	1.146	1.131	1.164	1.391	1.379	1.354
*Ho model*									
q_e,calc_ (mg·g^−1^)	253.13	256.88	262.92	246.05	251.24	253.98	249.82	266.95	276.84
k_2_ × 10^5^ (g·mg^−1^·min^−1^)	4.11	5.12	6.16	4.09	4.88	5.82	5.89	6.21	6.89
R^2^	0.990	0.989	0.990	0.991	0.988	0.988	0.987	0.993	0.993
χ^2^	76.336	89.173	79.617	62.628	92.856	88.002	93.383	63.624	59.873
*Elovich model*									
α (mg·g^−1^·min^−1^)	3.77	5.22	7.27	3.55	4.61	6.14	6.44	7.12	8.58
β (g·mg^−1^)	0.02	0.02	0.02	0.01	0.01	0.01	0.02	0.02	0.02
R^2^	0.967	0.960	0.958	0.971	0.959	0.957	0.954	0.962	0.964
χ^2^	247.758	325.663	347.937	212.145	312.975	338.627	352.700	319.802	331.668
*Weber-Morris intraparticle diffusion model*									
k_id_ (mg·g^−1^·min^−0.5^)	1.36	1.41	1.56	0.70	0.72	0.86	1.63	1.73	1.81
C_i2_	34.60	49.61	52.27	31.29	46.44	51.45	40.37	52.30	57.03
R^2^	0.995	0.995	0.995	0.994	0.994	0.995	0.995	0.995	0.996

**Table 12 polymers-14-04151-t012:** Thermodynamic parameters.

Sample Code	Δ*H* (kJ·mol^−1^)	Δ*S* (J·mol^−1^)	Δ*G* (kJ·mol^−1^)			*R* ^2^
			25 °C	30 °C	35 °C	
AE	19.81	80.28	−4.10	−4.50	−4.90	0.999
AD	18.36	73.31	−3.48	−3.85	−4.21	0.997
AT	29.17	113.21	−4.56	−5.12	−5.69	0.999
AEHA	24.34	107.07	−7.56	−8.09	−8.63	0.997
ADHA	19.40	86.42	−6.34	−6.77	−7.21	0.998
ATHA	30.82	129.53	−7.77	−8.42	−9.07	0.999

**Table 13 polymers-14-04151-t013:** Kinetic release parameters of metronidazole pH = 1.2.

Sample Code	Higuchi Model		Korsmeyer–Peppas Model			Baker–Lansdale Model	
	*k_H_* (min-^0.5^)	*R* ^2^	*k_r_* (min-^n^)	*n*	*R* ^2^	*K_BL_*	*R* ^2^
AE	4.101	0.991	0.082	0.40	0.998	0.047	0.990
AD	3.879	0.990	0.075	0.35	0.997	0.040	0.991
AT	4.258	0.994	0.087	0.42	0.998	0.051	0.989
AEHA	3.052	0.989	0.062	0.61	0.997	0.041	0.989
ADHA	2.736	0.990	0.037	0.57	0.998	0.034	0.988
ATHA	3.208	0.989	0.075	0.63	0.999	0.046	0.989

## Data Availability

The data presented in this study are available on request from the corresponding author.

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
