# Peer review of "Grafted Microparticles Based on Glycidyl Methacrylate, Hydroxyethyl Methacrylate and Sodium Hyaluronate: Synthesis, Characterization, Adsorption and Release Studies of Metronidazole"

_polymers, 2022, doi:10.3390/polym14194151_

Round 1

Reviewer 1 Report

Comments

Summary

This article has described the fabrication of “Three series of porous microparticles based on GMA, HEMA and one of the following crosslinking agents: EGDMA, DGDMA and TEGDMA by aqueous suspension polymerization. The microparticles are further surface modified by grafting sodium hyaluronate (HA) on the microparticles surface to introduce biocompatibility. Synthesis of the microparticles are well supported by different characterizations techniques.  After that loading and release of the antimicrobial drug (metronidiazole) was performed, which are supported by different models. Finally, Authors have claimed its potentiality as a drug delivery carrier for curing dental diseases such as periodontal disease.”   

Considering the contribution of this work in the field of drug delivery, this article can be accepted in “Polymers” after a major revision. Some of my major concerns are enlisted below-

Major Comments

1.       Some of the Authors of this manuscript have already published an article named “Complex microparticulate systems based on glycidyl methacrylate and xanthan.” Back in 2014 in the journal “Carbohydrate polymers”. Authors should explain what the exact difference between these two works is and why Authors have found HA is superior over xanthane for this work? Reference is mentioned below-

“Lungan, M. A., Popa, M., Desbrieres, J., Racovita, S., & Vasiliu, S. (2014). Complex microparticulate systems based on glycidyl methacrylate and xanthan. Carbohydrate polymers, 104, 213-222.”

2.       There are many polymers have been used to develop microgels and related delivery of metronidiazole for periodontitis. One of the examples is Pluronic® F127. Author should explain how their constructed work is superior from the other literature.

3.       Although the morphology of the microparticles are well observed from SEM analysis, AFM images does not fully corroborates the SEM study. Author should provide a clear AFM image with height profile to support their claim.

4.       Modify Figure 2, as an orange circle does not clearly reflects the surface modifications of the microparticles with HA.

5.       Authors should re-think the Title of this manuscript as this work mostly focus on the different model studies to support the adsorption of the drug at different conditions and its subsequent release study.  

6.       FTIR images are not clear (particularly wavenumbers are not visible). Please provide clear images.

Minor Comments

1.    Authors have used many odd words while constructing a sentence, for e.g. Line No- 230 “In vitro drug release studies were realized as follows:”. “Realized” can be replaced with “carried out” or “observed”. Rectify these kinds of words.

Author Response

Response to Reviewer

The authors of the paper entitled “Grafted microparticles based on glycidyl methacrylate, hydroxyl methacrylate and sodium hyaluronate for metronidazole release” are grateful to the reviewer for his valuable suggestions/recommendations. All the comments and suggestions have been very seriously taken under consideration and a careful revised manuscript has been accordingly prepared. All changes made following point-by-point the reviewer recommendations are marked in the manuscript with Track changes and are presented in the list included below.

Reviewer 1

Q1 Some of the Authors of this manuscript have already published an article named “Complex microparticulate systems based on glycidyl methacrylate and xanthan.” Back in 2014 in the journal “Carbohydrate polymers”. Authors should explain what the exact difference between these two works is and why Authors have found HA is superior over xanthane for this work? Reference is mentioned below-“Lungan, M. A., Popa, M., Desbrieres, J., Racovita, S., & Vasiliu, S. (2014). Complex microparticulate systems based on glycidyl methacrylate and xanthan. Carbohydrate polymers, 104, 213-222.”

Response: The work published in Carbohydrate Polymer journal is different from that submitted to the Polymers journal. The differences are:

  1. xanthan-grafted micropaticles are obtained in a single step more precisely directly from the suspension polymerization reaction;
  2. the monomers used in the organic phase are: glycidyl methacrylate and dimethacrylic monomers (EGDMA, DEGDMA, TEGDMA);
  3. the porogenic agent is toluene;
  4. the aqueous phase consists of the stabilizing agent (only poly(vinyl alcohol), xanthan, distilled water and a second initiator (ammonium persulfate). The second initiator was added to form macroradicals on the xanthan chains.

In the present work:

-   the preparation of microparticles took place in two stages;

- three monomers were used: glycidyl methacrylate, hydroxyethyl methacrylate and dimethacrylic monomers;

- the porogenic agent was n-butyl acetate;

- the aqueous phase consist of stabilizing agent (poly(vinyl alcohol), gelatin and sodium chloride) and distilled water.

Thus, the structures of both types of microparticles (published in Carbohydrate Polymer and from this manuscript) are different but may have some similar behaviors. The choice of sodium hyaluronate was explained in the introduction as follows “The choice of sodium hyaluronate, a derivative of hyaluronic acid as a natural component in the production of hybrid microparticles was based on the following considerations: hyaluronic acid is an essential component of the periodontal ligament matrix; it plays important roles in cell migration, adhesion and differentiation by binding proteins and cell receptors and has been studied as a metabolite or diagnostic marker of inflammation in gingival crevicular fluid” Also, the natural polymer has been elected to improve the biocompatibility of the microparticles.

Q2 There are many polymers have been used to develop microgels and related delivery of metronidiazole for periodontitis. One of the examples is Pluronic® F127. Author should explain how their constructed work is superior from the other literature.

Response: It is true that over time, several types of systems capable of encapsulating and releasing metronidazole for the treatment of periodontitis or other diseases have been developed, most of them being hydrogels/microgels, which use Pluronic® F127 as a carrier. It has the ability to micellize, loading into the core of the micelle mostly hydrophobic drugs, [I. Jarak, et al., Eur J Med Chem, Nov 15;206:112526, 2020, doi: 10.1016/j.ejmech.2020.112526. Epub 2020 Jul 17], e.g. curcumin for treating vaginal conditions or breast cancer [A. Kulkarni et al, Colloids Surf B Biointerfaces, 2021 Sep;205:111834. doi: 10.1016/j.colsurfb.2021.111834. Epub 2021 May 11]. For the treatment of periodontitis, metronidazole was loaded in complex gels based on Pluronic (which confers thermosensitivity) but mixed with methylcellulose and silk fibroin. Obviously, metronidazole which is a water soluble drug was encapsulated in the hydrogel network and not in the hydrophobic core of the Pluronic micelles [D.T. Pham et al ., 2021Turkish Journal of Pharmaceutical Sciences, January 18(4), 2020, doi : 10.4274/tjps.galenos.2020.09623]. In our case, we designed porous microparticles as carriers, capable of loading higher amounts of metronidazole both in the three-dimensional network, and especially by adsorption in the pores, the latter conferring a high specific surface area to the carriers. Their advantage consists, on the one hand, in the possibility of loading higher quantities of metronidazole, but also in the longer release time (more than 10 hours) compared to Pluronic microgels [max. 60 min, see ;  A. Kulkarni et al, Colloids Surf B Biointerfaces, 2021 Sep;205:111834. doi: 10.1016/j.colsurfb.2021.111834. Epub 2021 May 11].

Q3. Although the morphology of the microparticles are well observed from SEM analysis, AFM images does not fully corroborates the SEM study. Author should provide a clear AFM image with height profile to support their claim.

Response:

The reviewer is right. For better image clarity the AFM images of AE and AEHA microparticles with height profile were added.

Q4. Modify Figure 2, as an orange circle does not clearly reflects the surface modifications of the microparticles with HA.

Response: According to the reviewer’s good suggestion, the Figure 2 has been modified.

Q5. Authors should re-think the Title of this manuscript as this work mostly focus on the different model studies to support the adsorption of the drug at different conditions and its subsequent release study.

Response: As reviewer suggestion, the title of the manuscript has been changed in “Grafted microparticles based on glycidyl methacrylate, hydroxyethyl methacrylate and sodium hyaluronate: Synthesis, characterization, adsorption and release studies of metronidazole

Q6. FTIR images are not clear (particularly wavenumbers are not visible). Please provide clear images.

Response: The FTIR images were improved. For a better presentation and visualization, the spectra of AT and ATHA microparticles are included in the manuscript, while the other two are placed in the Supporting Information as Figure S1 and S2, respectively.

Q7.  Authors have used many odd words while constructing a sentence, for e.g. Line No- 230 “In vitro drug release studies were realized as follows:”. “Realized” can be replaced with “carried out” or “observed”. Rectify these kinds of words.

Response: The words have been corrected.

We thank the kind reviewer for his thoughtful comments that greatly improved the manuscript.

Reviewer 2 Report

This manuscript reports grafted microparticles based on glycidyl methacrylate, hydroxyethyl methacrylate and sodium hyaluronate for metronidazole release. The prepared hybrid microparticles present higher specific surface area, higher swelling capacities and higher adsorption capacities of metronidazole. The release mechanism of metronidazole in the case of hybrid microparticles is a complex mechanism being characteristic of anomalous or non-Fickian diffusion. The results are very good and interesting. However, some points of the manuscript should be improved. Specific comments are given below.

1.    The authors should revise the manuscript due to many sentences used as a paragraph.

2.    Diameters of precursor/hybrid microparticles in Table 2 should be consistent with the diameters in Figure 7. Size statistical distribution of SEM micrographs of hybrid microparticles should be provided.

3.    The structure of precursor/hybrid microparticles should be destructed which is due to the ester group in the structure. More discussion or data should be offered.

4.    Figure 12 is suggested to add deviation.

5.    The release mechanism of metronidazole of hybrid microparticles should be compared with other microparticles in previous reports.

Author Response

Response to Reviewer

The authors of the paper entitled “Grafted microparticles based on glycidyl methacrylate, hydroxyl methacrylate and sodium hyaluronate for metronidazole release” are grateful to the reviewer for his valuable suggestions/recommendations. All the comments and suggestions have been very seriously taken under consideration and a careful revised manuscript has been accordingly prepared. All changes made following point-by-point the reviewer recommendations are marked in the manuscript with Track changes and are presented in the list included below.

Reviewer 2

Q1 The authors should revise the manuscript due to many sentences used as a paragraph.

Response: According to the reviewer 2 suggestion the manuscript has been revised and some paragraphs were removed.

Q2 Diameters of precursor/hybrid microparticles in Table 2 should be consistent with the diameters in Figure 7. Size statistical distribution of SEM micrographs of hybrid microparticles should be provided.

Response: The differences between the size of microparticles from SEM images and Table 2 are due to the fact that the SEM images are made on a small number of particles, while the average diameters from Table 2 was carried out on larger number of microparticles (50 microparticles). Consequently, we have included in Supporting Information a new Figure S3 showing the microparticle size distributions determined by using a Laser Diffraction Particle Size Analyzer WingSALD 7001.

Q3 The structure of precursor/hybrid microparticles should be destructed which is due to the ester group in the structure. More discussion or data should be offered.

Response: The values of the parameter n from Korsmeyer-Peppas equation are less than 0.85, leading to the conclusion that the metronidazole release is achieved due to the swelling of the microparticles without any degradation or erosion of microparticles. In the manuscript the following sentence was added: “The values of n parameter are less than 0.85 leading to the conclusion that the microparticles swelled but did not undergo any disintegration or erosion”.

Q4 Figure 12 is suggested to add deviation. 

Response: The deviation was added to the Figure 12.

Q5 The release mechanism of metronidazole of hybrid microparticles should be compared with other microparticles in previous reports.

Response: The release mechanism of metronidazole from hybrid microparticles was compared with some microparticles presented in literature. For this purpose, the following paragraph was added: “Similar results have been found in the literature for other microparticulate systems. For example, in case of microparticles based on gelatin and poly(ethylene glycol) coated with ethyl cellulose, the metronidazole release rates and transport parameters suggesting the non-Fickian mechanism [43]. Also, the release kinetics of metronidazole from hydrogel containing crosslinked chitosan microparticles was best fitted to Higuchi model [44]”. In addition, two new references were added.

We thank the kind reviewer for his thoughtful comments that greatly improved the manuscript.

Round 2

Reviewer 1 Report

Congratulations! I am accepting this manuscript in its current form.